# The Ginsenoside Compound K Suppresses Stem-Cell-like Properties and Colorectal Cancer Metastasis by Targeting Hypoxia-Driven Nur77-Akt Feed-Forward Signaling

**DOI:** 10.3390/cancers15010024

**Published:** 2022-12-20

**Authors:** Minda Zhang, Zeyu Shi, Shuaishuai Zhang, Xudan Li, Sally Kit Yan To, Yijia Peng, Jie Liu, Siming Chen, Hongyu Hu, Alice Sze Tsai Wong, Jin-Zhang Zeng

**Affiliations:** 1State Key Laboratory of Cellular Stress Biology, Fujian Provincial Key Laboratory of Innovative Drug Target Research, School of Pharmaceutical Sciences, Faculty of Medicine and Life Sciences, Xiamen University, Xiamen 361102, China; 2School of Biological Sciences, University of Hong Kong, Pokfulam Road, Hong Kong SAR 999077, China; 3Xingzhi College, Zhejiang Normal University, Lanxi 321004, China

**Keywords:** compound K, orphan nuclear receptor Nur77, PI3K/Akt inhibitors, microRNA, colorectal cancer, cancer stem cells, drug target identification, molecular pharmacology

## Abstract

**Simple Summary:**

Compound K (CK) is the major functional ginsenoside metabolite derived from ginseng, one of the most popular traditional Chinese medicines, with unknown mechanistic targets. We describe here that CK directly binds and modulates Nur77 phosphorylation to deplete cancer stem-cell-like cells (CSCs) and suppress colorectal cancer (CRC) metastasis. Mechanistically, CK disrupts an unprecedented Nur77-Akt feed-forward loop to inactivate PI3K/Akt, which promotes and maintains the properties of CSCs under a hypoxic microenvironment. We thus identify Nur77 as the direct target of CK and dissect a novel mechanism for its action in CRC.

**Abstract:**

Hypoxia reprograms cancer stem cells. Nur77, an orphan nuclear receptor, highly expresses and facilitates colorectal cancer (CRC) stemness and metastasis under a hypoxic microenvironment. However, safe and effective small molecules that target Nur77 for CSC depletion remain unexplored. Here, we report our identification of the ginsenoside compound K (CK) as a new ligand of Nur77. CK strongly inhibits hypoxia-induced CRC sphere formation and CSC phenotypes in a Nur77-dependent manner. Hypoxia induces an intriguing Nur77-Akt feed-forward loop, resulting in reinforced PI3K/Akt signaling that is druggable by targeting Nur77. CK directly binds and modulates Nur77 phosphorylation to block the Nur77-Akt activation loop by disassociating Nur77 from the p63-bound Dicer promoter. The transcription of Dicer that is silenced under a hypoxia microenvironment is thus reactivated by CK. Consequently, the expression and processing capability of microRNA let-7i-5p are significantly increased, which targets PIK3CA mRNA for decay. The in vivo results showed that CK suppresses cancer stemness and metastasis without causing significant adverse effects. Given that the majority of FDA-approved and currently clinically tested PI3K/Akt inhibitors are reversible ATP-competitive kinase antagonists, targeting Nur77 for PI3K/Akt inactivation may provide an alternative strategy to overcoming concerns about drug selectivity and safety. The mechanistic target identification provides a basis for exploring CK as a promising nutraceutical against CRC.

## 1. Introduction

Colorectal cancer (CRC) ranks the third most common (1.85 million new cases per year) and the second deadliest cancer (850,000 deaths per year) worldwide [1,2]. Of all newly diagnosed patients, 20% have metastatic lesions when first diagnosed and another 25% with primary tumors will eventually develop metastases [3]. Currently available targeted drugs against metastatic CRC include cetuximab (Erbitux, anti-EGFR) [4], ramucirumab (Cyramza, anti-VEGR2) [5], regorafenib (Stivarga, small inhibitor against VEGFR) [6], and zaltrap (ziv-aflibercept, anti-VEGFA/B) [7,8]. However, many patients develop severe chemoresistance during the course of treatment [9]. Adverse side effects have been even reported in 79.36% of participants (173/218) [10]. Furthermore, angiogenesis inhibitors induce a hypoxic niche, thereby supporting tumor initiation by generating cancer stem cells (CSCs) [11,12]. Thus, although the last two decades have witnessed a significantly improved survival of CRC patients, metastatic diseases remain a great challenge.

CSCs, a rare population of cells within the heterogeneous CRC, are a major source of cancer relapse and metastasis [13]. Unlike bulk tumor cells, CSCs are drug-resistant and possess the capacity to differentiate and self-renew [14]. The recurrence after remission is likely due to the inability of current chemotherapeutic approaches to kill the CSCs and their capability to regenerate a new tumor [15]. Thus, understanding the key signaling pathways in CSC maintenance may lead to the identification of promising therapeutic targets. Hypoxia is pivotal for promoting and maintaining the plasticity of CSCs. Up to 50–60% of solid tumors consist of hypoxic regions [16,17], which have been shown to overlap with CSC niches [18]. In CRC, hypoxia can protect CSCs from anticancer drugs by downregulating the expression of MDR-1/ABCG2 and diminishing the efficiency of ionizing radiation therapy [19,20]. High levels of HIF-1α correlate with poor clinical outcomes in CRC patients [21]. However, whether the hypoxia-mediated molecular signaling axis acts as a therapeutic target in colorectal CSCs remains poorly understood. Moreover, current HIF-1 inhibitors are observed with side effects in normal tissues.

We recently reported that the orphan nuclear receptor Nur77 (also known as NR4A1, TR3, or NGFIB), which is extensively induced under a hypoxic microenvironment and can be activated by HIF-1α [22,23], is involved in the hypoxia-mediated enhancement of colorectal CSC properties [24], suggesting that Nur77 may potentially be a colorectal CSC-depleting target. It has been demonstrated that Nur77 is increased in a majority of CRCs and its overexpression is closely associated with advanced CRC stages, distant metastases, and poor patient survival [25,26]. Thus, targeting Nur77 is encouragingly a prospective approach for metastatic CRC therapy.

*Panax ginseng* C.A. Meyer has been used extensively as a panacea for thousands of years in China [27,28]. It is one of the most prevalent alternative medicines in the world and appears in the pharmacopeias of China, the U.S., and Europe for its purported beneficial effects with no adverse side effects [29]. Compound K (CK; (20-O-(β-D-glucopyranosyl)-20(S)-protopanaxadiol) is the major functional metabolite of ginsenosides [30,31]. Thus, unraveling its molecular mechanisms of action is essential for developing it as a nutraceutical. 

Here, we show for the first time that CK directly binds Nur77 to suppress Nur77-mediated stemness and metastasis in vitro and in vivo via disrupting Nur77-Akt feed-forward signaling under a hypoxic microenvironment, revealing a new target and novel mechanism of action for CK for the treatment of metastatic CRC.

## 2. Materials and Methods

### 2.1. Antibodies and Reagents

Anti-Nur77 (#3960), anti-Dicer (#5362), and anti-phospho-Akt (#4060) were purchased from Cell Signaling Technology (Danvers, MA, USA); anti-phospho-serine/threonine (ab15556) from Abcam (Cambridge, MA, USA); anti-p63 (12143-1-AP), anti-CD44 (15675-1-AP), anti-CD133 (18470-1-AP), anti-EpCAM (21050-1-AP), anti-ALDH-1 (15910-1-AP), and anti-β-actin (66009-1-Ig) from Proteintech (Rosemont, Inc., Rosemont, CA, USA); anti-total Akt (sc-81434), anti-p110α (sc-293172), anti-GFP (sc-9996), anti-Flag (sc-166355) and anti-c-Myc (sc-40) from Santa Cruz Biotechnology (Santa Cruz, CA, USA). Lipofectamine 2000 from Invitrogen (Carlsbad, CA, USA); Compound K from AMCEC (Shanghai, China); MK-2206 from Beyotime (Shanghai, China); and fetal bovine serum (FBS) from Gibco (Grand Island, NY, USA). 

### 2.2. Molecular Docking

Autodock 4.2. was used to analyze the docked complex between CK (conformation from Pubchem at https://pubchem.ncbi.nlm.nih.gov/, accessed on 26 January 2021) and Nur77 (the X-ray crystal structure was retrieved from Protein Data Bank with a PDB code of 4JGV). The grid center was set at positions of −12.08 (X-axis), −18.29 (Y-axis), and −4.233 (Z-axis) with a grid box size of 40 × 40 × 40 (0.375 Å), as described previously [32]. PyMOL v1.7.4 was used for molecular visualization.

### 2.3. Isothermal Titration Calorimetry (ITC)

The thermodynamic parameters of the interaction between CK and Nur77 were determined with MicroCal 2000 microcalorimeter (GE Healthcare, Little Chalfont, UK). The purified Nur77/LBD protein was pooled to a concentration of 50 μM in 1X PBS containing 0.1% Triton X-100 (pH 7.4) at 25 °C. CK was prepared as a stock of 1 mM in the same buffer in a syringe and then automatically titrated into a Nur77/LBD-containing sample pool at a constant velocity for 40 min. ITC data were analyzed using ITC programs (OriginPro 2019b v9.6.5).

### 2.4. Fluorescence Titration

Fluorescence-quenching measurements were performed by titrating 2 mL of purified Nur77/LBD (1 μM) in a quartz cuvette with 20 drops of 1 mM CK. The samples were excited at 280 nm and the fluorescence emission was captured from 300 to 500 nm. The fluorescence spectrums were collected with a fluorescence spectrophotometer (Varian Eclipse, Palo Alto, CA, USA) and analyzed with OriginPro 2019b v9.6.5.

### 2.5. Cell Culture and Transfection

HCT116, SW480, and HEK293T cell lines were purchased from American Type Culture Collection (ATCC; Manassas, VA, USA). HCT116 was cultured in RPMI-1640 medium (Gibco), while SW480 and HEK293T were maintained in Dulbecco’s Modified Eagle Medium (DMEM) (Gibco). Both the cultured media contained 10% FBS. The cells were grown in a CO_2_ incubator at 37 °C with 5% CO_2_. A hypoxic cell incubator (1% O_2_) was used to mimic a hypoxic microenvironment. All the cells were cultured under a hypoxia microenvironment during CK treatment unless indicated. Cell transfection was performed with Lipofectamine 2000 (Invitrogen) according to the manufacturer’s instructions. 

### 2.6. Co-Immunoprecipitation

Cells were lysed in 500 μL of Co-IP lysis buffer (50 mM HEPES, pH 7.4; 100 mM NaCl; 10% glycerol; 1% triton X-10; 25 mM NaF) at 4 °C for 1h. The lysates were incubated with specific antibodies at 4 °C overnight and purified with 40 μL of protein A/G sepharose (Santa Cruz Biotechnology). The beads were washed twice with Co-IP lysis buffer and analyzed by Western blotting.

### 2.7. Western Blotting

The prepared protein samples were separated on 8% SDS-PAGE gels and electro-transferred to a PVDF membrane (Millipore, Billerica, MA, USA). The membrane was blocked in 5% milk followed by incubation with primary antibodies at 4 °C overnight. After washing three times, the membrane was then incubated with horseradish peroxidase-conjugated secondary antibodies at room temperature for 2 h and visualized with enhanced chemiluminescence detection reagents (Yeasen, Shanghai, China). 

### 2.8. Real-Time PCR and qPCR

Total RNA was extracted by Trizol and reversely transcribed into cDNA using the HiScript^®^ II Q Select RT kit (Vazyme, Nanjing, China); 2×Hieff^®^ PCR Master Mix (Yeasen) and Hieff^®^ qPCR SYBR Green Master Mix (Yeasen) were used for RT-PCR and qPCR assays, respectively. The specific primers used include: CD44: 5′-CTG CCG CTT TGC AGG TGT A-3′ (forward) and 5′-CAT TGT GGG CAA GGT GCT ATT-3′ (reverse); CD133: 5′-AGT CGG AAA CTG GCA GAT AGC-3’ (forward) and 5’-GGT AGT GTT GTA CTG GGC CAA T-3′ (reverse); Dicer: 5′-TGC TAT GTC GCC TTG AAT GTT-3′ (forward) and 5′-AAT TTC TCG ATA GGG GTG GTC TA-3′ (reverse); PI3KCA: 5′-GTA TGT CTA TCC GCC ACA TGT AG-3′ (forward) and 5′-CAC AGT CAT GGT TGA TTT TCA GAG-3′ (reverse); and β-actin: 5′-CTC CAT CCT GGC CTC GCT GT-3′ (forward) and GCT GTC ACC TTC ACC GTT CC (reverse). To detect miRNA levels, the purified RNA sample was first reversely transcribed to cDNA by TransScript^®^ miRNA First-Strand cDNA Synthesis Kit (TransGen, Beijing, China) and then amplified by qPCR [24]. The miRNA data were analyzed by the 2-ΔΔCT method.

### 2.9. Tumor Sphere Formation Assay

CRC cells were cultured as tumor spheres for enriching CSC from a serum starvation system [24]. Briefly, 1000 cells/well were seeded in 24-well ultra-low attachment plates (Corning, Corning, NY, USA) in serum-free medium containing 20 ng/mL human epidermal growth factor (Invitrogen), 1×B27 (Invitrogen), 20 ng/mL basic fibroblast growth factor (Invitrogen), and 20 ng/mL IGF (Invitrogen). After 2 weeks, the tumor spheres were imaged by light microscopy; the number of spheres with a diameter of >50 μm was counted.

### 2.10. CRISPR/Cas9

The CRISPR/Cas9 system was used to knock out Nur77, p63, and Dicer in CRC cells. The target sequences of sgRNA were designed with the online tool (http://crispr.mit.edu, accessed on 28 December 2020) and cloned into the LentiCRISPR v2 (Addgene plasmid #52961). The lentivirus harboring interested sgRNA was then used to generate stable lines. Briefly, HEK293T cells were transfected with transfer plasmid, packaging plasmid psPAX2 (Addgene, #12260), and envelope plasmid VsVg (Addgene, #8454) for 48 h. The supernatant was harvested and used to infect CRC cells. Stable cell lines were selected with puroMycin followed by selecting the monoclonals for further culture. The efficiency of infection was tested by Western blotting. The specific primers used include: sgControl (targeting a non-genic region): 5′-CAC CGT CGC GCT CGG CGG GTC ACA G-3′ (forward) and 5′-AAA CCT GTG ACC CGC CGA GCG CGA C-3′ (reverse); sgNur77: 5′-CAC CGT CCG AAC AGA CAG CCT GAA G-3′ (forward) and 5′-AAA CCT TCA GGC TGT CTG TTC GGA C-3′ (reverse); sgp63: 5′-CAC CGA GGG TCA GGG CAG TAC TGT A-3′ (forward) and 5′-AAA CTA CAG TAC TGC CCT GAC CCT C-3′ (reverse); and sgDicer: 5′-CAC CGA GTC CAA AGA AAG GAC CCA T-3′ (forward) and 5′-AAA CAT GGG TCC TTT CTT TGG ACT C-3′ (reverse).

### 2.11. Dual-Luciferase Reporter Assays

Dicer promoter luciferase reporter and renilla luciferase plasmids were transfected into CRC cells. The cells were lysed in a reporter lysis buffer (Beyotime). Dicer promoter luciferase reporter activities were measured by Multiskan Spectrum (PerkinElmer, Waltham, MA, USA) and normalized to renilla fluorescence intensity. 

### 2.12. Chromatin Immunoprecipitation Assay

Cells were crosslinked with 1% formaldehyde at 37 °C for 10 min and then sonicated in lysis buffer (50 mM HEPES-KOH, 150 mM NaCl, 1 mM Na_2_EDTA·2H_2_O, 1% Triton X-100). The lysates were incubated with anti-p63 antibody or rabbit IgG at 4 °C overnight. After incubation with A/G agarose beads at 4 °C for 2 h, 5 M of NaCl was added to reverse the cross-linking at 65 °C for 4 h. The immunocomplexes were digested with proteinase K at 55 °C for 3 h. The DNA was extracted by phenol/chloroform and precipitated by ethanol. The purified chromatin was subjected to PCR. The primers for amplifying the Dicer promoter (from −1584 bp to −1478 bp) were: 5′-TGG GAA GCC GAG GCA GGT CA-3′ (forward) and 5′-AGG CAT GTG CCA CCA CGC CC-3′ (reverse) [24]. 

### 2.13. Animal Experiments

BALB/c nude mice (6 weeks old) purchased from SLAC Laboratory Animal Co., Ltd. (Shanghai, China) were housed and maintained in Xiamen University Animal Experiment Centre. A lung metastatic model for human CRC was developed by tail vein injection with HCT116 cells (2 × 10^6^ cells/mouse). For convenient in vivo imaging and observation, the cells were stably expressed with a luciferase reporter. After one month of vein injection, the mice were randomized into four groups (*n* = 6 for each group) based on the bioluminescence intensities with the IVIS imaging system. The mice were treated intragastrically once daily with vehicle or CK (25 mg/kg or 50 mg/kg) for 4 weeks. During the treatment, the mice were weighed and the metastatic lesions were imaged weekly by injection intraperitoneally with 150 mg/kg D-luciferin. To determine the impact of Nur77 phosphorylation on CRC metastasis, HCT116 Nur77/KO cell lines were stably transfected with Nur77/wt, Nur77/S351A, or Nur77/S351D, which were similarly injected into mice tail veins. The mice were grown for 2 months to allow metastatic nodule development. At the end of the experiment, the mice were killed. The metastatic nodules in the lung were calculated. The nodules striped from lung tissues were stored at −80 °C for Western blotting. The lung and other organ tissues including the heart, liver, kidney, and spleen were collected and fixed in 4% paraformaldehyde for hematoxylin-eosin (H&E) staining. All animal experiments were approved by the Xiamen University Animal Care and Use Committee.

### 2.14. Statistical Analysis

All data reported in this study were representative of three independent experiments and presented as means ± standard deviation (SD). The statistical significance of differences was compared by Student’s *t*-test between two groups or by one-way ANOVA for more than two groups. *p* < 0.05 was considered to be statistically significant.

## 3. Results

### 3.1. CK as a New Ligand for Nur77

Compound K (CK; (20-O-(β-D-glucopyranosyl)-20(S)-protopanaxadiol, Figure 1A), which is converted in the intestine from ginsenosides by gut microbiota, was shown to possess pharmacological activities against various human diseases via diverse pathways [33]. However, its direct target remains completely unknown. In our pilot study, we found that CK could modulate Nur77 expression and suppress CRC growth. We thus attempted to determine whether CK could bind and regulate Nur77 function. Autodock-based virtual screening showed that Nur77 harbors a putative CK-binding site on the hydrophobic, grooved surface of its ligand-binding domain (LBD). The sugar moiety of CK is suggested to form four hydrogen bonds with the polar residues of Thr576, Glu445, Leu506, and Leu556 (Figure 1A). We then used fluorescence titration and isothermal titration calorimetry (ITC) to identify the potential binding. Our results showed that CK could dose-dependently bind Nur77/LBD with a KD of 15.55 μM (Figure 1B,C). To confirm, the predicted binding sites were mutated. When the leucine at position 556 was replaced with tryptophan or glutamic residue at 445 with alanine, the KD values soared from 15.55 μM to 1.11 mM and 2.74 mM, respectively. Replacing Leu506 with tryptophan also significantly interfered with the binding affinity (KD = 265.96 μM), while mutating Thr576 into alanine did not (KD = 17.67 μM) (Figure 1D). Thus, we demonstrated that CK is a new ligand for Nur77. 

### 3.2. CK Inhibits CSC Phenotypes Dependent on Nur77

We recently reported that the hypoxic induction of Nur77 is involved in promoting colorectal CSC phenotypes [24]. We thus explored whether CK could deplete CSCs by targeting Nur77. For this purpose, a 3D spheroid culture system was established. HCT116 and SW480 cells were grown under stem-cell-selective conditions and treated with increasing concentrations of CK (0, 5, 10 μM). Our results showed that CK could dose-dependently inhibit sphere formation, as evidenced by smaller foci and fewer spheres by CK in HCT116 (30.6% at 5 μM and 50.6% at 10 μM) and SW480 (35.3% at 5 μM and 57.8% at 10 μM) than their corresponding vehicle controls (Figure 2A). Since Nur77 was implicated in the regulation of several CSC markers such as CD44, CD133, EpCAM, and ALDH-1 [24], their expression levels were compared between CK and vehicle controls. The results showed that 10 μM CK elicited significant drops in the mRNA levels of CD44 (−74.5%), CD133 (−79.2%), EpCAM (−38.7%), and ALDH-1 (−54.8%) in HCT116 (Figure 2B, left panel). Similar extents of inhibition were seen in SW480 (Figure 2B, right panel). The significant inhibitory effect of CK on the transcription of CD44, CD133, EpCAM, and ALDH-1 was also confirmed by qPCR (Figure 2C). Consistently, the protein levels of CD44, CD133, EpCAM, and ALDH-1 were reduced in a CK-dose-dependent manner (Figure 2C,D). Such an effect was efficiently blocked when Nur77 was knocked out by CRISPR/Cas9 (Appendix A). To explore whether the binding is critical for CK action, the Nur77/KO cells were reintroduced with the wild-type (wt), L556W, or E445A of Nur77. The stable cells were again exposed to CK treatment. The results showed that the effects of CK on inhibiting sphere formation and CSC phenotypes (CD44 and CD133) were reproducible in wt cells, but lost in both L556W- and E445A-transfected cells (Figure 2F,G and Appendix A). Together, these results demonstrate that Nur77 is critical for CK-mediated CSC suppression in hypoxic CRC.

### 3.3. Nur77 Is Hyper-Phosphorylated under a Hypoxic Microenvironment

Interestingly, hypoxia extensively induced Nur77 phosphorylation, as detected by anti-Ser/Thr antibodies in both endogenously and ectopically expressed proteins (Figure 3A,B). Nur77 phosphorylation upon hypoxic stimuli could be reversed by MK-2006, a specific inhibitor against Akt, indicating Nur77 as a substrate for phosphorylation by Akt under a hypoxic microenvironment (Figure 3A,B). The biological significance of Akt-mediated Nur77 phosphorylation, which remains completely unknown, was then investigated. Since Nur77 possesses a putative Akt phosphorylation site at position Ser351 [34,35,36], we mutated this site with alanine (S351A, an unphosphorylatable form) or aspartic residue (S351D, a constitutively phosphorylated form). The Nur77/KO (sgNur77) cells were re-expressed stably with the wild-type (Nur77/wt) or mutated form of Nur77/S351A or Nur77/S351D. When growing in a 3D culture environment, the overexpression of Nur77/wt and Nur77/S351D greatly promoted CSC sphere growth (Figure 3C), which was associated with the increased expression of CD44 and CD133 (Figure 3D). Consistently, Nur77/wt and Nur77/S351D strongly stimulated CRC metastasis (Figure 3E,F) and CSC phenotypes (Figure 3G). In contrast, the transfection of Nur77/S351A did not induce significant changes in the sphere formation, metastasis, and CSC phenotypes (Figure 3C–G). These results suggest that Nur77 phosphorylation at S351 is implicated in the CSC formation and metastasis of CRCs. 

### 3.4. Akt-Mediated Phosphorylation Regulates Nur77 Interaction with p63

Since our previous study showed that Nur77-dependent CSC phenotypes were mediated by transgressing p63 transcriptional activity [24], we explored whether Akt-mediated Nur77 phosphorylation could modulate Nur77/p63 interaction. Indeed, Nur77 dephosphorylation by MK-2006 or calf intestinal phosphatase (CIP) strongly suppressed its interaction with p63 (Figure 4B). Consistently, the interaction between Nur77 and p63 was blocked or enhanced when the serine at 351 was mutated to alanine (S351A) or aspartic residue (S351D) (Figure 4C). The overexpression of Akt promoted p63 to interact with Nur77/wt but not with Nur77/S351A (Figure 4D,E). The untunable phosphorylation form of Nur77/S351D could constitutively interact with p63, which could not be affected by MK-2206 (Figure 4F). Our in vivo experiments further confirmed that the Ser351 phosphorylation of Nur77 regulated its interaction with p63 (Figure 4G). 

### 3.5. CK Disrupts Hypoxia-Induced Nur77 Complex with p63 on Dicer Promoter by Modulating Nur77 Phosphorylation

Nur77 phosphorylation induced by hypoxia or the overexpression of Akt could be inhibited by CK treatment (Figure 5A,B). CK-mediated Nur77 dephosphorylation strongly impaired its interaction with p63 (Figure 5C and Appendix A). In contrast, CK did not interfere with the interaction between Nur77/S351D and p63 (Appendix A). In agreement with the critical role of p63 in Dicer transcription [24], CK treatment potently promoted Dicer promoter luciferase activity (Figure 5D) and enhanced p63 to bind the Dicer promoter, as indicated in the ChIP assay (Figure 5E). Consistently, Dicer mRNA and protein levels were upregulated by CK (Appendix A). The occupancy of p63 on the Dicer promoter could be regulated by Nur77 upon its phosphorylation status, as indicated in Nur77/wt- and Nur77/S351D- rather than Nur77/S351A-transfected cells (Figure 5F), which was also confirmed by in vivo analysis (Figure 5G). In the presence of CK, the binding of p63 on the Dicer promoter was recovered in Nur77/wt but not Nur77/S351D-transfected cells (Figure 5F). Consistently, CK greatly rescued the protein levels of Dicer downregulated by Nur77/wt, while Nur77/S351D-mediated Dicer suppression was resistant to CK regulation (Figure 5H). Dicer expression was upregulated by CK in a p63-dependent manner, as evidenced in p63 knockout cells (Figure 5I). Taken together, we showed that CK induces Dicer expression by suppressing Nur77 S351 phosphorylation, which prevents Nur77 from inactivating the transcriptional activity of p63.

### 3.6. Depletion of CSC Markers by CK Is Associated with the Disruption of the Nur77-Akt Feed-Forward Loop

PI3KCA is critical for CSC maintenance under a hypoxia microenvironment [24]. CK could dose-dependently inhibit PI3KCA mRNA expression (Figure 6A), followed by the reduced protein expression of p110α and the dephosphorylation of Akt (Figure 6B). CK-mediated p110α downregulation and Akt inactivation could be reversed by Dicer/KO (sgDicer) (Figure 6B,C). Let-7i-5p was shown to regulate PI3KCA mRNA stability [24]. We showed here that CK could strongly promote pre-Let-7i processing (Appendix A) and increase Let-7i-5p expression (Appendix A). Such an effect was abrogated when Dicer was knocked out (Appendix A). Thus, Dicer and Let-7i-5p mediated the effect of CK on PI3KCA expression and Akt activation. 

Intriguingly, hypoxia induced Nur77-dependent Akt activation, which in turn upregulated Nur77 phosphorylation to form Nur77-Akt feed-forward signaling. To support this, in the absence of CK, the overexpression of Nur77/wt and Nur77/S351D could strongly upregulate p110α expression and Akt phosphorylation, while Nur77/S351A did not (Figure 6C). Upon CK treatment, p110α expression and Akt phosphorylation were downregulated in Nur77/wt- but not Nur77/S351D-transfected cells (Figure 6D,E). Consistently, the hypoxic enhancement of CSC phenotypes and sphere formation was inhibited by CK in a manner dependent on a tunable phosphorylation status of Nur77 at S351 (Figure 6D,E). Taken together, these data suggest that CK could disrupt the Nur77-Akt feed-forward loop by modulating Nur77 phosphorylation.

### 3.7. The Anti-Metastasis Effect of CK In Vivo

BALB/c nude mice were intravenously implanted with HCT116 cells that expressed a luciferase reporter for the in vivo monitoring of metastatic nodule growth via the Caliper IVIS Lumina System (Waltham, MA, USA) (tail-vein injection metastasis model). The fluorescent images of metastatic lesions were captured weekly (Figure 7A). The results showed that CK treatment (25 mg/kg and 50 mg/kg) could dose-dependently inhibit lung metastasis, as evidenced in fluorescent images (Figure 7A,B), metastatic nodule calculation (Figure 7C,D), and H&E staining (Figure 7E). Consistent with our in vitro results, the administration of CK led to a reduction in p-Akt, p110α, CD44, and CD133 in the metastatic nodules (Figure 7F). Thus, a schematic mechanism for CK to target the Nur77-Akt loop was proposed (Figure 7G). CK appeared to be capable of preventing weight loss in comparison with the untreated mice, who were heavily burdened with metastatic lesions (Appendix A). CK did not cause histological alterations in normal tissues of the heart, liver, lungs, spleen, and kidney when compared to vehicle-injected mice (Appendix A). Further, CK did not significantly increase the serum levels of alkaline phosphatase (ALP), alanine transaminase (ALT), creatinine, or blood urea nitrogen (BUN) (Appendix A). These results suggest that CK is safe when used to inactivate the PI3K/Akt pathway for metastatic suppression in mice. 

## 4. Discussion

Ginseng has been a distinguished Chinese traditional medicine to treat various human diseases for thousands of years. After being metabolized by the intestinal microbiota, PPD-type ginsenosides would then be converted into CK, which shows superior bioavailability among other ginsenosides and their derivatives [37]. In this study, we show for the first time the effects of CK on cancer cells experiencing hypoxia. The treatment of hypoxic CRC cells with CK significantly blocks Nur77-mediated oncogenic signaling, leading to the suppression of CSC properties and metastases in vitro and in vivo with no apparent adverse effects, suggesting that CK could be a safe and effective therapeutic. 

Most recently, studies have revealed that dysbiosis of the gut microbiota might be an essential contributor to the initiation and progression of CRC [38]. Some bacteria, such as *Streptococcus gallolyticu*, *Fusobacterium nucleatum*, *Escherichia coli*, *Bacteroides fragilis*, and *Enterococcus faecalis*, have a high prevalence in CRC patients as compared to the normal population [39,40,41,42,43], whereas *Clostridiales, Oscillibacter, Ruminococcus, Holdemania*, and *Sutterella* are generally required for the biotransformation of Rb1 to CK [44]. The mismatch of microbiota species between CRC patients and CK transformation indicates an indispensable potential benefit of directly using CK compared to its parental compounds. Thus, our identification of CK may serve as a promising lead in treating CRC. 

Hypoxia is a prominent feature of solid tumors and correlates with the poor prognosis of cancer patients. Tumor cells undergo a stemness transformation in hypoxia [45], in turn causing relapse at local or distant sites, suggesting that hypoxia is highly associated with a more aggressive tumor phenotype [46]. CSCs primarily grow in hypoxic niches and hypoxia acts as a powerful driving force to maintain and promote characteristic and tumorigenic CSCs. Thus, attention to the contrapose hypoxic tumor microenvironment in treating CSC cannot be ignored. Interestingly, Nur77 is evidenced as a critical regulator of CSCs under a hypoxic microenvironment [24]. Our previous study reported that the hypoxic activation of PI3K/Akt signaling accompanied by increased colorectal CSCs is dependent on Nur77 induction and its interaction with p63 [24]. Intriguingly, we further show here that hypoxia-induced Nur77 phosphorylation at Ser351 mediated by Akt regulates its interaction with p63 (Figure 4), which forms a feed-forward loop of Nur77-Akt, resulting in reinforced PI3K/Akt signaling. Nur77 plays both tumor-promoting and suppressing roles that are largely dependent on its phosphorylation status and subcellular localization. Specific phosphorylation modulation may change the function and role of Nur77. Nur77 phosphorylation by JNK is essential for inducing its nuclear export and apoptotic effect [35], while phosphorylation by ERK and Pin1 enhances its oncogenic potential [47,48]. Nur77 is overexpressed and hyper-phosphorylated in most CRCs [49]. Nur77 plays a paradoxical role in CRCs, which is at least partially due to its different phosphorylation modifications [50]. The Akt-mediated phosphorylation of Nur77 impairs its DNA binding activity and nuclear export [34]. The tumor-promoting role of Nur77 largely relies on its DNA binding and transcriptional activity, while its nuclear export is tumor-suppressive. Thus, Akt-mediated phosphorylation provokes a complex effect on the function of Nur77. The biological significance of Nur77 by Akt remains completely unknown. We show here that the reciprocal regulation of Nur77 and Akt under hypoxia results in reinforced PI3K/Akt signaling and enhanced CSC properties. Importantly, this Nur77-Akt feedback loop is druggable, as shown in the CK treatment of CRC in vitro and in vivo. The CK ligation of Nur77 was accompanied by the depletion of CSCs and the suppression of CRC metastasis. Mechanistically, when bound by CK, Nur77 displays resistance to Akt phosphorylation and is dissociated from p63, causing disruption to the Nur77-Akt feed-forward loop (Figure 7G). Introducing mutations into CK binding sites on Nur77 renders CK to lose its capability to inactivate Akt.

The hyperactivation of PI3K has been found in about 50% of human malignancies. In CRC, this rate is even as high as 60–70% [51], in which 20–30% is attributable to the PIK3CA mutation, which codes for the mutated active form of class IA PI3K catalytic subunit p110α. Studies have suggested that the interaction between hypoxia and Akt is bilateral, with the activation of either one increasing another, and p110α acts as a key resistance promoter in CSCs during this process. The PI3K/Akt signaling is indispensable to the development of CRC and drug resistance in CSCs [52]. Given the important roles of the PI3K/Akt pathway in CSC formation during the progression of tumor initiation and relapse, developing therapeutic inhibitors against PI3K may be a promising therapeutic strategy to eliminate CSCs in cancer therapy. However, the majority of FDA-approved and currently clinically tested PI3K inhibitors are ATP-competitive reversible kinase antagonists, raising concerns about drug selectivity and safety. Our findings suggest that targeting the hypoxic Nur77-Akt feed-forward loop is anticipated to have higher drug selectivity and safety than currently available PI3K/Akt inhibitors.

## 5. Conclusions

Together, we provide mechanistic evidence supporting a role for CK to target Nur77 in eliminating hypoxic CSCs for the first time, offering a promising novel approach with the use of plant-derived small molecule compounds for CRC patients.

## Figures and Tables

**Figure 1 cancers-15-00024-f001:**
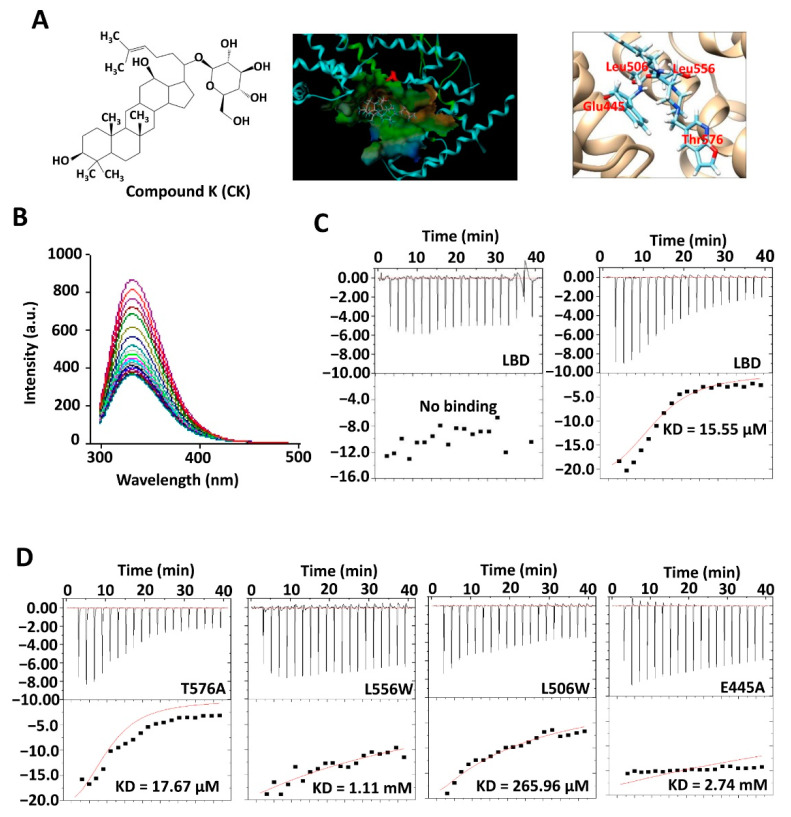
CK is a new ligand for Nur77. (**A**) Chemical structure of CK (left), docking CK into a potential binding pocket (middle) with predicted binding sites of Nur77 based on Autodock virtual analysis (right). (**B**) Fluorescence titration assay. Titrate 1 μM Nur77/LBD protein in a quartz cuvette with 20 drops of 1 mM CK. The fluorescence was captured and analyzed with sample excitation spectra at 280 nm and emission spectra from 300 to 500 nm. (**C**,**D**) ITC assay. A sample pool of 50 μM Nur77/LBD (**C**) or various mutants as indicated (**D**) was titrated with vehicle (left) or 1 mM CK (right) automatically for 40 min. The affinity of CK for Nur77 is expressed as a KD value.

**Figure 2 cancers-15-00024-f002:**
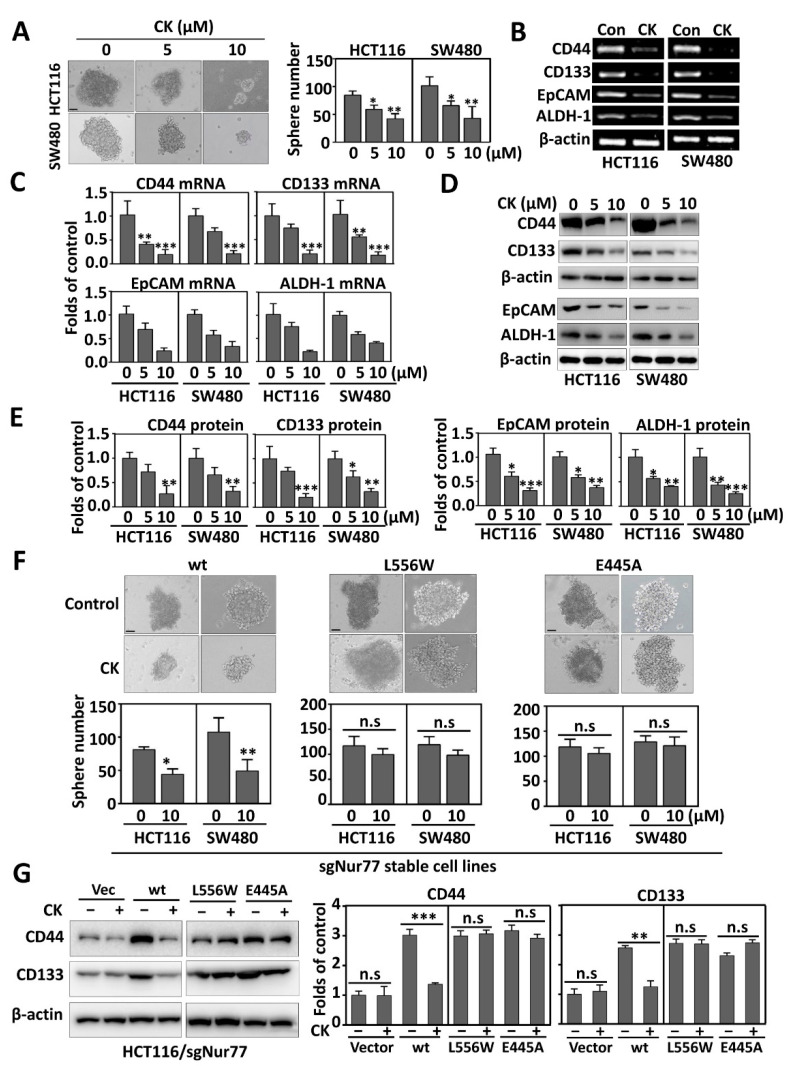
CK inhibits CSC phenotypes dependent on Nur77. (**A**) CRC cells were cultured at 1% O_2_ in serum-free CSC-selective medium and treated with CK (0, 5, or 10 μM) every 48 h for 2 weeks. Spheres (>50 μm) were photographed (left) and counted (right). (Scale bar, 50 μm). (**B**) CRC cells were cultured at 1% O_2_ and treated with vehicle or 10 μM CK for 24 h. The mRNA expression levels of CSC markers were examined by RT-PCR. (**C**) CRC cells were exposed to CK (0, 5, or 10 μM) for 24 h under 1% O_2_. The RNA was purified for qPCR assay. (**D**) CRC cells were cultured and treated with CK in the same way as that described in (**C**). The lysates were subjected to a Western blotting assay. (**E**) Quantification of protein levels based on bot grey intensities in (**D**). (**F**,**G**) The endogenous Nur77 in CRC cells was knocked out by CRISPR/Cas9 and then stably re-expressed with Nur77/wt, Nur77/L556W, or Nur77/E445A. The cells were cultured as tumor spheres with similar treatment and analysis as described in (**A**) (**F**) or grown in RPMI-1640 medium with 24 h treatment of CK (0 or 10 μM) followed by assaying the changes in CSC makers. (Scale bar, 50 μm). (**G**). * *p* < 0.05; ** *p* < 0.01; *** *p* < 0.001 vs. respective control. n.s.= non-significant. Molecular weights for proteins are indicated in the full, uncropped, annotated Western blot images (Appendix A).

**Figure 3 cancers-15-00024-f003:**
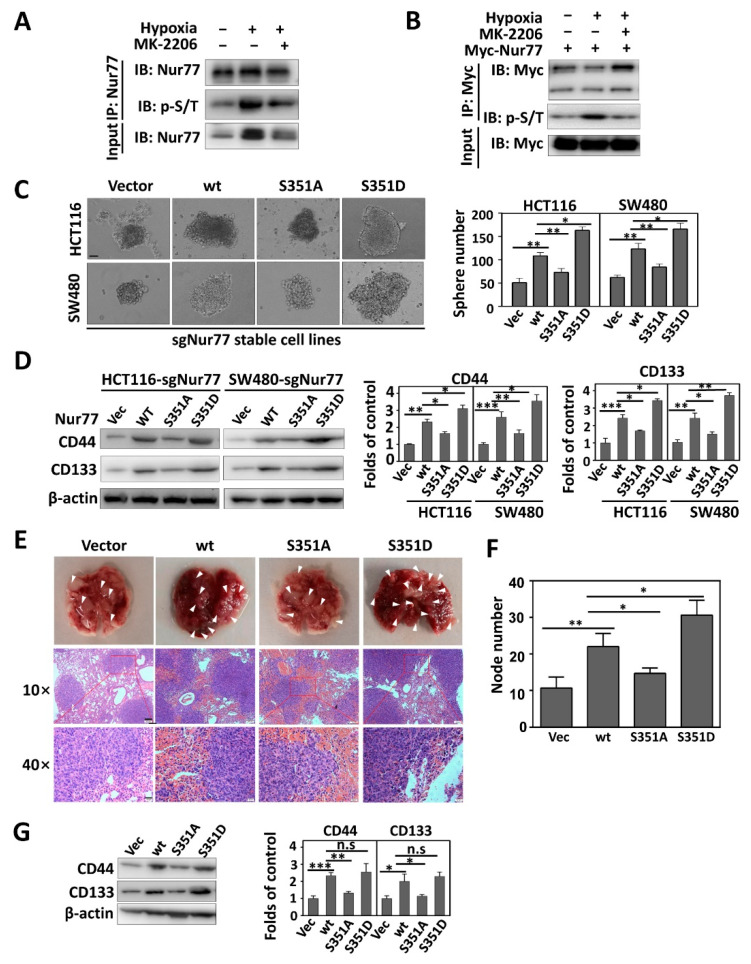
Nur77 is hyper-phosphorylated under a hypoxic microenvironment. (**A**,**B**) Hypoxic stimuli. HCT116 and its stable cell line expressing Myc-Nur77 were cultured under normoxia (20% O_2_) or hypoxia (1% O_2_), followed by treatment with or without 1 μM MK-2206 for 4 h. The cells were lysed and immunoprecipitated with anti-Nur77 antibody (**A**) or anti-Myc antibody (**B**) and blotted with p-Ser/Thr (p-S/T) antibody. (**C**,**D**) Implication of Nur77 phosphorylation in CSC phenotypes. The CRC cells that carried vector, Nur77/wt, or mutant at the phosphorylation site of S351 (S351A or S351D) were cultured as tumor spheres in a 3D environment for 2 weeks; spheres (>50 μm) were photographed and counted (scale bar, 50 μm) (**C**). Otherwise, the cells were grown in a normal medium under hypoxia for 24 h (**D**). The effects of Nur77 mutation at S351 on CSC markers (CD44 and CD133) were determined by Western blotting (**D**, left). The blot grey intensities were quantified (**D**, right). (**E**–**G**) BALB/c nude mice were subjected to the tail-vein injection of HCT116 Nur77/KO cells stably transfected with vector, Nur77/wt, Nur77/S351A, or Nur77/S351D (*n* = 6 per group) and allowed to develop metastatic nodules (about 2 months). Representative metastatic nodules are shown (upper panel). The tumors were also subjected to H&E staining (lower panel). Scale bars: 100 μm in 10× and 20 μm in 40×. The white arrows represented the metastatic nodules. (**E**). The numbers of metastatic nodules were statistically analyzed (**F**). Metastatic tissues were lysed and blotted for detecting CD44 and CD133 expression (**G**). * *p* < 0.05; ** *p* < 0.01; *** *p* < 0.001 vs. respective control. n.s.= no significant. Molecular weights for proteins are indicated in the full, uncropped, annotated Western blot images (Appendix A).

**Figure 4 cancers-15-00024-f004:**
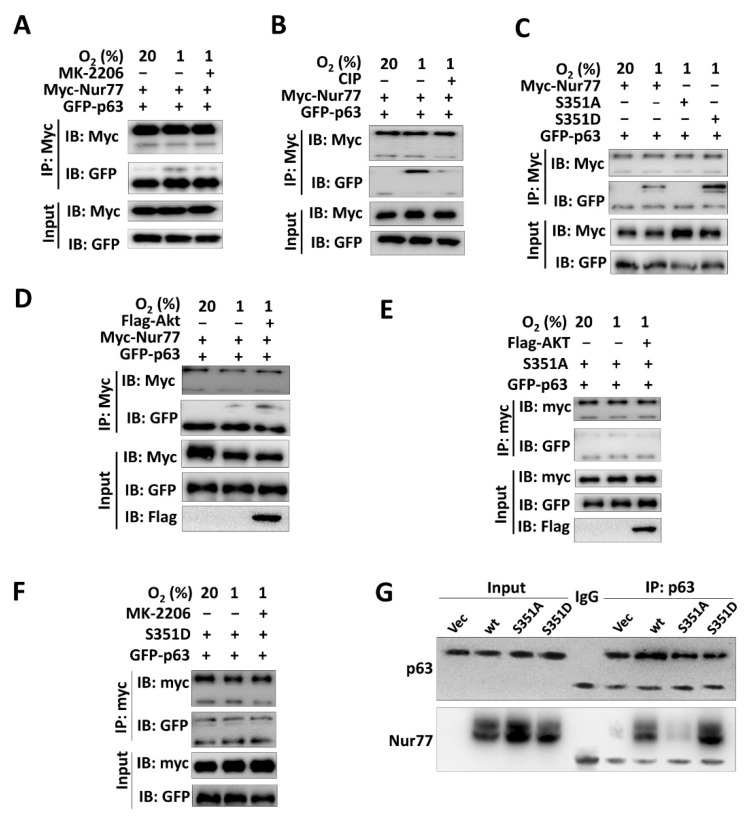
Akt-mediated phosphorylation regulates Nur77 interaction with p63. HEK-293T cells carried by different expression vectors were used. (**A**,**B**) The cells were cultured under normoxia (20%) or hypoxia (1%). Hypoxic cells were treated with or without 1 μM MK-2206 for 4 h (**A**). The lysates collected from untreated cells under hypoxia were incubated with or without CIP (0.2 U/mL) at 37 °C for 60 min (**B**). (**C**) Mutations were introduced into the Nur77 phosphorylation site of S351. (**D**,**E**) The overexpression of Akt modulated the interaction between Nur77 (or mutant) and p63. (**F**) The interaction of Nur77/S351D (untunable phosphorylation form) with p63 was analyzed in HEK-293T cells following the administration of vehicle or 1 μM MK-2206 (Akt inhibitor). (**G**) The metastatic tissues were collected from a tail-vein metastatic model, as indicated in Figure 3E. (**C**–**G**) the capability of Nur77 or its mutants to interact with p63 was evaluated with Co-IP assays. Molecular weights for proteins are indicated in the full, uncropped, annotated Western blot images (Appendix A).

**Figure 5 cancers-15-00024-f005:**
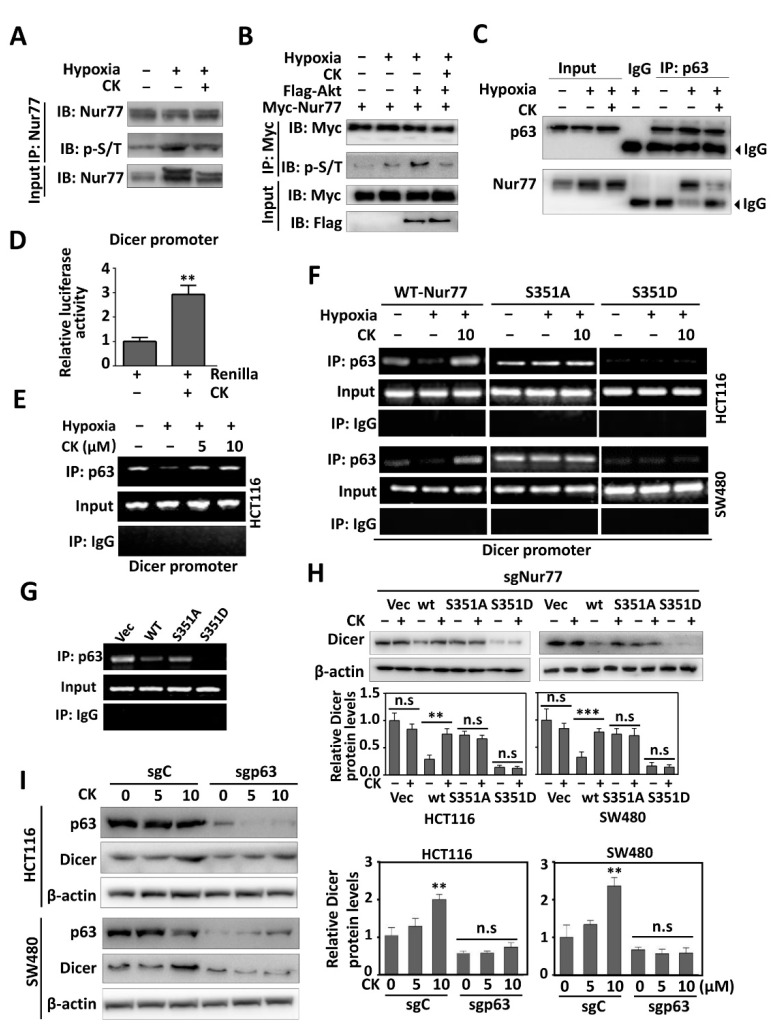
CK disrupts hypoxia-induced Nur77 interaction with p63 on the Dicer promoter by modulating Nur77 phosphorylation. (**A**,**B**) Untransfected and transfected HCT116 cells were cultured under normoxia or hypoxia. Hypoxic cells were treated with or without 10 μM CK for 4 h. The lysates were immunoprecipitated with anti-Nur77 (**A**) or anti-Myc (**B**). The phosphorylation of Nur77 was examined with p-Ser/Thr (p-S/T) antibody. (**C**) The endogenous interaction between Nur77 and p63 was disrupted when HCT116 cells were exposed to 10 μM CK treatment for 24 h. The cell lysates were immunoprecipitated with anti-p63 and blotted with antibodies against Nur77 and p63. (**D**) Reporter assay. HCT116 cells were co-transfected with Dicer promoter plasmid and renilla luciferase for 24 h and then exposed to 24 h of treatment with vehicle or 10 μM CK. The luciferase activities were measured by Multiskan Spectrum (PerkinElmer, USA). The renilla fluorescence intensity was used to normalize for transfection efficiency. (**E**,**F**) HCT116 and its transfected cells with indicated expressing vectors were cultured under normoxia or hypoxia. Hypoxic cells were treated with vehicle, 5 μM, or 10 μM CK for 24 h. Cell lysates were immunoprecipitated with anti-p63 antibody and analyzed using specific Dicer promoter primers (ChIP assay). Nonspecific IgG was used as a negative control. (**G**) The metastatic tissues collected as indicated in Figure 3E were immunoprecipitated with anti-p63 antibody and subjected to the ChIP assay, as described above. (**H**) CRC cells harboring different expression vectors were treated with or without 10 μM CK for 24 h. (**I**) p63/WT and p63/KO cells were treated with vehicle, 5 μM, or 10 μM CK for 24 h. For H and I, the levels of Dicer protein were detected and quantified by Western blotting. ** *p* < 0.01; *** *p* < 0.001 vs. respective control. n.s. = non-significant. sgC = sgControl. Molecular weights for proteins are indicated in the full, uncropped, annotated Western blot images (Appendix A).

**Figure 6 cancers-15-00024-f006:**
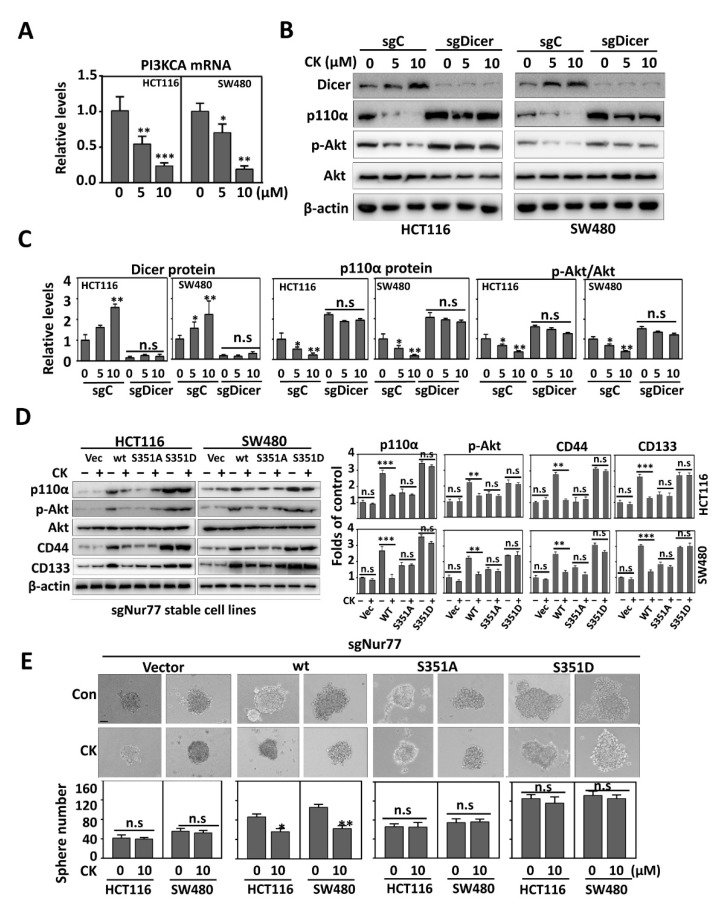
The depletion of CSC markers by CK is associated with Nur77-mediated Akt inactivation. (**A**) CRC cells were treated with vehicle, 5 μM, or 10 μM CK for 24 h under hypoxic conditions. The mRNA level of PIK3CA was examined by qPCR. (**B**) Dicer/WT and Dicer/KO cells were treated with vehicle, 5 μM, or 10 μM CK for 24 h. (**C**) The quantification of protein levels based on bot grey intensities in (**B**). (**D**) Nur77/KO cells were transfected with indicated Nur77 mutants and treated with or without 10 μM CK for 24 h. For (**B**,**D**), the protein expression levels were examined by Western blotting. (**E**) Nur77/KO cells were transfected with indicated plasmids and cultured as tumor spheres in the presence or absence of 10 μM CK. Spheres (> 50 μm) were photographed and counted (scale bar, 50 μm). * *p* < 0.05; ** *p* < 0.01; *** *p* < 0.001 vs. respective control. n.s. = non-significant. sgC = sgControl. Molecular weights for proteins are indicated in the full, uncropped, annotated Western blot images (Appendix A).

**Figure 7 cancers-15-00024-f007:**
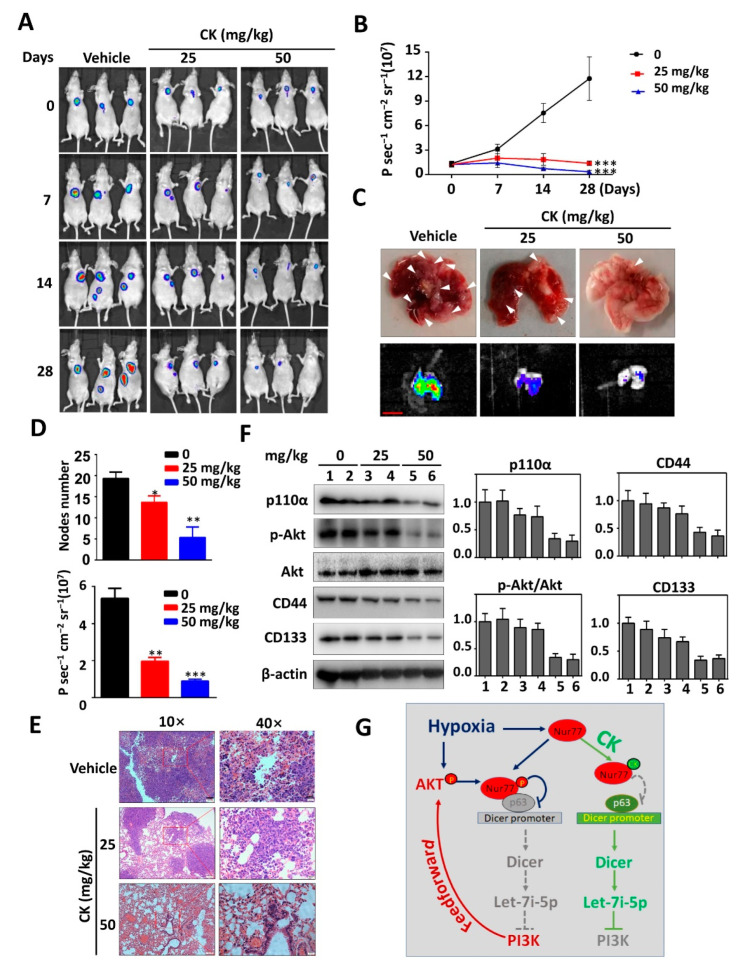
The anti-metastasis effect of CK in vivo. (**A**,**B**) BALB/c nude mice were injected with HCT116-luciferase cells through tail veins. After one month of CRC cell injections, the animals were treated with vehicle and CK (25 mg/kg and 50 mg/kg) once daily for 28 days. Bioluminescence images (**A**) and intensities (**B**) were obtained from the IVIS Lumina system every week. (**C**) Representative metastatic nodule images in the lungs of mice after 28 days of treatment. Scale bar, 1 cm (**D**) The numbers of metastatic nodules were statistically analyzed. (**E**) Representative lung H&E staining image. Scale bars: 100 μm in 10× and 20 μm in 40×. (**F**) Metastatic nodules collected from each group were lysed and analyzed by Western blotting for the detection of p110α, p-Akt, Akt, CD44, and CD133. (**G**) A proposed schematic mechanism. * *p* < 0.05, ** *p* < 0.01; *** *p* < 0.001 vs. respective control. Molecular weights for proteins are indicated in the full, uncropped, annotated Western blot images (Appendix A).

## Data Availability

The data presented in this study are available within the article or Appendix A.

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
