# Peer review of "The Ginsenoside Compound K Suppresses Stem-Cell-like Properties and Colorectal Cancer Metastasis by Targeting Hypoxia-Driven Nur77-Akt Feed-Forward Signaling"

_cancers, 2022, doi:10.3390/cancers15010024_

Round 1

Reviewer 1 Report

This manuscript described that the major functional ginsenoside metabolite CK directly binds and modulates Nur77 phosphorylation to deplete CSCs and suppress colorectal cancer metastasis through disrupting Nur77-Akt feedforward loop under hypoxic microenvironment. It is interesting and novel. However, there are some questions to be addressed.

1. In Figure2D, the authors detected the CD44 and CD133 protein expression in CRC cells. To be consistent with C, the EpCAM and ALDH-1 protein expression should be detected using WB because both EpCAM and ALDH-1 are important cancer stem cell markers.

2. Is there only one phosphorylation site at position Ser351 by Akt kinase for Nur77 according to reference 34? Or there may be an undiscovered phosphorylation site.

3. In Figure3E-F, the number of metastatic lung cancer nodules should include both internal and external nodules in lung, and the total number should be larger, which also seen in Figure 7 B-C. How exactly did the authors  calculate the number of metastatic lung cancer nodules.

4. In Figure 4, why does the authors don't use tumor cells (HCT116 or SW480) to detect the interaction of Nur77 and p63 instead of 293T cells.

5. It will be better to use a graphic abstract to summary the hypothesis of this manuscript. 

Author Response

1. In Figure2D, the authors detected the CD44 and CD133 protein expression in CRC cells. To be consistent with C, the EpCAM and ALDH-1 protein expression should be detected using WB because both EpCAM and ALDH-1 are important cancer stem cell markers.

Response: Thanks for the reviewer’s suggestion. We reloaded the samples and probed the protein expression of EpCAM and ALDH-1, which have now been provided in Figure 2D.

2. Is there only one phosphorylation site at position Ser351 by Akt kinase for Nur77 according to reference 34? Or there may be an undiscovered phosphorylation site.

Response: Akt-mediated Ser351 phosphorylation of Nur77 has also been described by others; the references have now been updated. We could not exclude that Akt may induce Nur77 phosphorylation at other sites. However, mutation of Ser351 strikingly affects Nur77 phosphorylation and its interaction with p63. We thus conclude that this phosphorylation site is critical for Nur77’s function.

3. In Figure3E-F, the number of metastatic lung cancer nodules should include both internal and external nodules in lung, and the total number should be larger, which also seen in Figure 7 B-C. How exactly did the authors calculate the number of metastatic lung cancer nodules?

Response: Thanks for the suggestion. We re-calculate the metastatic number to include both internal and external nodules. The data have now updated.

4. In Figure 4, why does the authors don't use tumor cells (HCT116 or SW480) to detect the interaction of Nur77 and p63 instead of 293T cells.

Response: We have used HCT116 cells to show that Nur77 could interact with p63 and the interaction was impaired upon CK treatment (Figure 5C). The interaction of Nur77 with p63 has also been determined in HCT116 xenograft tumor (Figure 4G). Since the cell line of 293T is highly transfectable and has been widely described in literatures to study protein-protein interaction. We thus also used 293T cells to determine the interaction of various mutant forms of Nur77 with p63 (Figure 4E-F). We hope that our explanation is acceptable.

 5. It will be better to use a graphic abstract to summary the hypothesis of this manuscript.

Response: Thanks for the suggestion. We have now provided a schematic mechanism in Figure 7F, while the Graphical Abstract has been submitted in a separated file.

Reviewer 2 Report

Zhang et al in this exhaustive study delineate the mechanisms by which compound K suppresses cancer stem cell properties in colorectal cancer cells. Using several arduous techniques, they demonstrate that CK targets Nur77-Akt loop without much adverse effects. It is impressive that they have addressed their hypothesis comprehensively. However, in addition to few suggestions for the experiments, there is a need for fixing the English language in the whole manuscript (including consistency of using words such as using hyphen between feed forward). I have listed my both major and minor concerns below:

1.       Line 27-28: remains unexplored

2.       Line 33: “Hypoxia induced Dicer silence” does not make sense.

3.       Line 235: shown to be

4.       In line 236, cite the study.

5.       Fog figure 1B and 1C briefly explain what was done in the results section.

6.       Line 264: cite the article

7.       Line 265: sphenoid?

8.       The effect seen on tumor spheroids: is this reduction in self-renewal or proliferation?

9.       Line 462: The use of word “tumor” interchangeably with metastatic nodules on the lung is misleading. Therefore, in 3.7 is the effect on metastasis or tumor formation?

10.   I strongly recommend including a schematic (like graphical abstract) explaining the observations.

Author Response

1. Line 27-28: remains unexplored

Response: Corrected now.

2. Line 33: “Hypoxia induced Dicer silence” does not make sense.

Response: Thanks for the reviewer’s comments. The sentence has now been improved.

3. Line 235: shown to be

Response: Corrected now.

4. In line 236, cite the study.

Response: Thanks for the suggestion. The reference has now been updated.

5. For figure 1B and 1C briefly explain what was done in the results section.

Response: Thanks for the suggestion. We have edited the result section in revised manuscript. 

6. Line 264: cite the article

Response: Thanks for the suggestion. The reference has now been updated.

7. Line 265: sphenoid?

Response: Corrected now.

8. The effect seen on tumor spheroids: is this reduction in self-renewal or proliferation?

Response: Yes, CK inhibits the self-renewal of CSCs, which has now been corrected.

9. Line 462: The use of word “tumor” interchangeably with metastatic nodules on the lung is misleading. Therefore, in 3.7 is the effect on metastasis or tumor formation?

Response: We agree with the reviewer’s comments. In 3.7, the effect of CK is to suppress metastasis rather new tumor formation, which has now been edited.

 10. I strongly recommend including a schematic (like graphical abstract) explaining the observations.

Response: Thanks for the suggestion. We have now provided a schematic figure in Figure 7F.